# Global and local ancestry estimation in a captive baboon colony

Christopher Kendall[ID][1]*, Jacqueline Robinson[2], Guilherme Debortoli[ID][3], Amin Nooranikhojasteh[4], Debbie Christian[5], Deborah Newman[ID][5], Kenneth Sayers[5], Shelley Cole[5], Esteban Parra[3], Michael Schillaci[6], Bence Viola[1]

1 Department of Anthropology, University of Toronto, Toronto, Ontario, Canada, 2 Institute for Human Genetics, University of California, San Francisco, San Francisco, California, United States of America, 3 Department of Anthropology, University of Toronto Mississauga, Mississauga, Ontario, Canada, 4 Epigenomics Lab, Princess Margaret Cancer Center, University Health Network, Toronto, Ontario, Canada, 5 Southwest National Primate Research Center, Texas Biomedical Research Institute, San Antonio, Texas, United States of America, 6 Department of Anthropology, University of Toronto Scarborough, Scarborough, Ontario, Canada

* chris.kendall@mail.utoronto.ca

**Data Availability Statement:** All VCFs, genetic map files, list of AIMs, and fixed marker list are available from Zenodo at https://doi.org/10.5281/zenodo.11176353.

## Abstract

The last couple of decades have highlighted the importance of studying hybridization, particularly among primate species, as it allows us to better understand our own evolutionary trajectory. Here, we report on genetic ancestry estimates using dense, full genome data from 881 olive (*Papio anubis*), yellow (*Papio cynocephalus*), or olive-yellow crossed captive baboons from the Southwest National Primate Research Center. We calculated global and local ancestry information, imputed low coverage genomes (n = 830) to improve marker quality, and updated the genetic resources of baboons available to assist future studies. We found evidence of historical admixture in some putatively purebred animals and identified errors within the Southwest National Primate Research Center pedigree. We also compared the outputs between two different phasing and imputation pipelines along with two different global ancestry estimation software. There was good agreement between the global ancestry estimation software, with $R^2 > 0.88$, while evidence of phase switch errors increased depending on what phasing and imputation pipeline was used. We also generated updated genetic maps and created a concise set of ancestry informative markers (n = 1,747) to accurately obtain global ancestry estimates.

## Introduction

Baboons (genus *Papio*) are Old World monkeys from the subfamily Cercopithecinae that have been classified into 6 currently identified species [1–3] or subspecies [4]. Generally speaking, genetic studies will refer to them as species, while morphological studies will discuss subspecies. For this paper, we will refer to them as species. These species include the olive (*P. anubis*), yellow (*P. cynocephalus*), hamadryas (*P. hamadryas*), kinda (*P. kindae*), guinea (*P. papio*), and chacma baboons (*P. ursinus*) [2, 3]. Early mitochondrial DNA evidence showed that *Papio spp.*

**Funding:** CK was supported by an NSERC CGS-D grant (CGSD2 - 535025 - 2019) for the duration of this research (https://www.nserc-crsng.gc.ca/index_eng.asp). The establishment, maintenance, and biological characterization of the pedigreed baboon colonies at the Southwest National Primate Research Center of Texas Biomedical Research Institute (SNPRC at Texas Biomed) has been supported in large part by grants to Texas Biomed Investigators by the National Institutes of Health (P51 RR013986, P01 HL028972) (https://www.nih.gov/) (DC, DN, KS, SC). The funders had no role in study design, data collection and analysis, decision to publish, or preparation of the manuscript. We would like to acknowledge NIH grant R24 OD017859 (https://www.nih.gov/) for their role in funding the sequencing of our samples. There was no additional external funding received for this study.

**Competing interests:** The authors have declared that no competing interests exist.

diverged into a northern and southern clade approximately 2.1 million years ago [1]. This was later investigated using whole genome data and this study found that baboons began radiating approximately 1.4 million years ago, where guinea, hamadryas, and olive baboons separated into a northern clade while chacma, kinda, and yellow baboons separated into a southern clade [2]. A recent, large study using 225 baboons with whole genome data, data from the X and Y chromosomes, and mitochondrial DNA supports the conclusion that baboons separated into northern and southern clades and identified kinda baboons as more basal than the other 5 extant species [3]. Similar to hominins, baboons evolved in savannah habitats in Africa [5, 6], and isotope studies have shown that baboons and early hominins, such as *Australopithecus africanus* and *Paranthropus robustus*, exploited similar dietary resources [7]. A recent review article emphasized the relevance of baboons for human evolutionary studies, as baboons and early hominids are thought to share similarities in environmental flexibility, body size, subsistence strategies, predator avoidance, and the size and structure of their societies [8].

Due to their similarity to humans, baboons have been studied in biomedical contexts. For instance, baboons have been used extensively in medical research [9, 10], including research on vaccines [11], osteoporosis [12, 13], obesity [14–17], and diabetes [18]. Much of this research has been conducted using animals from the Southwest National Primate Research Center (SNPRC), which maintains a colony of around 2,000 baboons. Their extensive breeding programs have generated a pedigree that includes over 16,000 individuals through seven generations, many of which have corresponding DNA or tissue samples available for research purposes [10]. The colony was established in the 1960s using presumed purebred olive and yellow baboons captured in Kenya [19] and has since grown substantially from the original 384 founders [10].

Studies on captive baboons have also been done to assess hybridization between olive and yellow baboons, with some studies using high coverage genomic data from the SNPRC population. Robinson et al. [19] analyzed a subset of 100 animals from the SNPRC, utilizing 33 founders and 67 captive-born baboons. Their results highlighted errors in the original SNPRC pedigree, showing that some individuals were admixed, despite being labeled as purebred, while other individuals were incorrectly labeled at the species level [19]. This agrees with more recent research done by Vilgalys et al. [20] based on 442 genomes from olive and yellow baboons in the Amboseli National Park in Kenya and 39 captive founders from the SNPRC and Washington National Primate Research Center. They showed that all of the baboons they sampled from Amboseli were admixed (mean 37% olive ancestry), and that yellow baboons used to start the SNPRC colony were actually admixed with upwards of 22% olive ancestry [20]. Previous research from Amboseli using 44 animals found that even putatively unadmixed yellow baboons still had large tracts of olive ancestry, upwards of 26%, suggesting a much more complex historical process that is more dynamic than what may appear based on just phenotypic data alone [21]. Additionally, Sørensen et al. [3] compiled 225 high coverage genomes from wild baboons across 19 different regions in Africa from the six different species and found evidence for both ancient and ongoing admixture.

Genetic studies on baboons have up until very recently been hampered by poor genetic characterization. The first publicly available baboon genome (*Pcyn1.0*) was of a yellow baboon from the SNPRC [21], however, this had small contig lengths and was fairly fragmented [22]. An olive baboon genome (*Panu_2.0*) was available for some time but it remained under embargo for many years [21] making it difficult to gain access and utilize. A third iteration of the baboon genome (*Panu_3.0*) was released recently, but like *Panu_2.0*, was still based on mapping contigs and scaffolds to the rhesus macaque (*Macaca mulatta*) genome, however, it did improve the assembly by closing gaps and removing several contigs [2]. A higher resolution olive baboon genome (*Panubis1.0*) was published soon after, that briefly, improved contig

lengths, increased the number of complete genes represented, and included the Y-chromosome [22]. A series of original baboon genetic maps were released previously but they contained only a small number of baboon markers [23, 24]. An updated genetic map based on linkage disequilibrium (LD) from 24 SNPRC founders was recently published for *Panu_2.0* [19], while an updated recombination map for *Panubis1.0* has also been published [25].

Here, we report the results from our analyses of whole genome sequencing data for 881 olive, yellow, and olive-yellow crossed baboons from the SNPRC, that were previously mapped to *Panubis 1.0*. To our knowledge, this is the largest database of high-resolution baboon genomic data to date. There is substantial variation in sequencing coverage for these samples. Consequently, for low-coverage samples, we performed imputation using high-coverage reference genomes. We then carried out global and local ancestry analyses and provided information on the underlying genetic structure within the colony. In addition, we produced updated genetic maps in both PLINK [26] and SHAPEIT5 [27] formats created from *Panubis1.0* information and provide imputed and phased VCF files containing 881 SNPRC animals. We also generated a panel of 1,747 ancestry informative markers (AIMs) that provide estimates of global ancestry that are highly correlated to genome-wide ancestry estimates along with over 27,000 markers that are fixed ($F_{ST} = 1$) between olive and yellow baboons. Our analyses provide further resources for this unique captive colony useful for future baboon genomic studies.

## Materials and methods

### Sequencing

In this study, we used publicly available whole genome sequencing data from 901 individuals from a pedigreed captive colony of putatively purebred olive and yellow baboons, and their crosses, at the SNPRC. All raw sequencing data used here are available from the NCBI Sequence Read Archive under BioProject PRJNA433868. Subsets of this complete dataset have been used in analyses for previously published studies [20, 25]. We briefly restate the methods for generating this dataset here.

Genomic DNA originally extracted from blood or tissue samples at the SNPRC were sent to MedGenome, Inc. for library preparation and 150-bp paired-end sequencing on HiSeq 2500, HiSeq 4000, or HiSeq X Ten machines (Illumina, San Diego, US). 756 individuals were sequenced to low coverage (median ~5X), and 145 samples were sequenced to high coverage (median ~35X). Raw reads were trimmed with TrimGalore v0.6.4 [28] to remove adapter contamination and low-quality bases (options: -q 20—stringency1—length 50) before being processed with a pipeline based on the Best Practices workflow for the Genome Analysis Toolkit v3 (GATK v3.8-1-0-gf15c1c3ef) [29]. Specifically, trimmed reads were first aligned to the *Panubis1.0* reference assembly (GCA_008728515.1) [22] with BWA MEM v0.7.17 [30]. Duplicate reads were then marked with Picard v2.21.3 [31], and genotyping was performed with GATK HaplotypeCaller (minimum base quality score 10, minimum mapping quality score 20) followed by GenotypeGVCFs and LeftAlignAndTrimVariants [29].

### Filtering

Raw VCF calls were hard filtered using BCFtools v1.13 [32] initially so only biallelic SNPs on autosomes were included. Repetitive elements for *Panubis1.0* were downloaded from both RepeatMasker [33] (data from: https://ftp.ncbi.nlm.nih.gov/genomes/all/GCF/008/728/515/GCF_008728515.1_Panubis1.0/) and Tandem Repeats Finder [34] (data from: https://hgdownload.soe.ucsc.edu/goldenPath/papAnu4/bigZips/). The corresponding *Panu_3.0* Tandem Repeats Finder file was lifted over to match the *Panubis1.0* coordinates using CrossMap v0.6.4 [35] and all repetitive elements were removed using BEDTools v2.30.0 [36]. Further

filtering on individual variants was done using the following parameters using BCFtools [32]: QD < 2, FS > 20, SOR > 2, MQ < 40, INFO/DP < 10, INFO/DP > 25, MQRankSum < -12.5, ReadPosRankSum < -8, QUAL < 30, FORMAT/GQ < 20, FORMAT/DP < 3, FORMAT/DP > 40, and ExcHet < 0.5. These filtering methods were based upon recommendations from the GATK hard-filtering protocol [37] and the filtering criteria from previous SNPRC analysis [19]. After sequencing, filtering, and removal of repetitive elements, we were left with 96,222,633 markers across the 901 samples. Lastly, we removed any samples not of olive, yellow, or olive-yellow cross ancestry according to the pedigree, leaving us with 881 samples for analysis.

## Creation of genetic map

We downloaded the recombination map published in Wall et al. [25] from the hosted data site (https://datadryad.org/stash/dataset/doi:10.7272/Q6HH6H9D). We used the formula of 1 centimorgan (cM) per megabase (1 million base pairs) (Mb) is equivalent to a $1e^{-8}$ recombination rate per base pair for conversion. To convert physical position to genetic position in cM we multiplied the physical length of the region (end coordinates subtracted from start coordinates) by the recombination rate per base pair and then multiplied by 100. We then used PLINK v1.9 [26] using the—recode feature with the—cm-map flag to create PLINK-style genetic maps compatible with Beagle 4.1 [38], Beagle 5.4 [39] (see Phasing, genotype refinement, and imputation of low-quality samples section) and RFMix v2.03-r0 [40] (see Global and local ancestry section). We also reformatted this map to be compatible with SHAPEIT5 v5.1.1 [27] and IMPUTE5 v1.1.5 [41] (see Phasing, genotype refinement, and imputation of low-quality samples section).

## Phasing, genotype refinement, and imputation of low-quality samples

We were interested in creating a dataset with as high confidence calls as possible. We used VCFtools v0.1.16 [42] to identify high coverage samples (> = 15X; which is 3X the low coverage mean) that were labelled either olive, yellow, or an olive-yellow cross in the pedigree. After identifying these individuals, we used ADMIXTURE v1.3.0 [43] in unsupervised mode with cross validation (—cv) activated with K = 1–10 to find the optimal number of populations found within the SNPRC sample. To do this, we extracted only the high coverage SNPRC founder individuals labeled as either olive or yellow from the pedigree (n = 51 total; n = 42 olive and n = 9 yellow) and then performed linkage disequilibrium (LD) pruning in PLINK [26] (—indep-pairwise 50 5 0.1) prior to ADMIXTURE [43] analysis, as suggested by the manual. Cross validation ended at K = 2, as expected, and we then used these individuals as a reference panel for phasing and imputation. We took the original filtered VCF file with over 96 million variants and ran additional filtering for missingness per marker (—geno 0.0) and minor allele frequency (MAF) (—maf 0.05) using PLINK [26] on only these 51, high-coverage animals, allowing us to retain 6,708,892 high quality markers. To create a baseline marker set from the low coverage individuals (those with <15X depth as reported in VCFtools [42]; n = 830) we modified the PLINK [26] filtering above to remove any markers with more than 5% missingness (—geno 0.05) and kept the MAF filtering the same (—maf 0.05). We relaxed the missingness parameter for the low coverage dataset as when we applied the same 0% missing filter (—geno 0.0) as we did with the high-coverage animals, we removed over 99% of the variants from the low coverage dataset. However, after relaxing this parameter in the low coverage samples, we were left with 5,640,014 markers, approximately a 16% reduction in the number of quality markers compared to the high coverage, no missing markers data set. We

then imputed the low coverage samples using the high coverage panel as references to improve the number of quality markers for analysis.

The remaining 830 samples first underwent genotype refinement, as suggested by Martin et al. [44] for handling of low coverage samples. The 51-sample reference panel was first phased using Beagle 5.4 [39] using standard settings and input parameters. Beagle 5.4 [39] was chosen due to its high accuracy and low amount of switch errors in a non-human population [45]. The low coverage sample genotypes were then refined using Beagle 4.1 [38] with the gtgl flag added to estimate posterior genotype probabilities with the 51 animals used as a reference panel. Beagle 5.4 [39] was used again to phase and impute the low coverage panel after refinement with the same 51 animal reference panel. BCFtools [32] was used to merge the imputed and reference VCFs, followed by filtering for only high confidence imputed calls, with any imputed markers with $DR^2$ values less than 0.7 being removed.

Based on analyses of putative olive/yellow first generation (F1) crosses in our study, there was evidence of phase switch errors (see Results). We attempted to remedy this by using Tractor, which implements local ancestry strategies to correct for phase switch errors, which they call "unkinking" [46]. After running Tractor [46], we were able to fix some of the switch errors found in our dataset. To further clean our data and to investigate whether the phasing issues were due to the software, we repeated our analysis using a second phasing and imputation pipeline, this time with SHAPEIT5 [27] and IMPUTE5 [41]. SHAPEIT5 [27] has the benefit of having an option to accept a pedigree file, and we speculated that this may decrease the number of switch errors. We ran SHAPEIT5 [27] with standard phase_common settings with the —pedigree flag engaged. We used the same 51 animal reference and 830 animal target panel for both phasing and imputation pipelines. The phased, low coverage dataset was imputed using IMPUTE5 [41] following the suggestion of breaking the files into smaller chunks per chromosome and then ligating them back together post-imputation to speed up computation time. Lastly, we merged the VCFs and filtered out imputed markers with INFO < 0.7 (IMPUTE5's version of $DR^2$) using BCFtools [32] and then reran Tractor [46].

## Global and local ancestry

Global ancestry was first estimated using ADMIXTURE [43]. As discussed above, an unsupervised ADMIXTURE [43] run was first completed solely on the 51 high coverage, founder individuals. Out of this initial set, only 26 (n = 17 olive and n = 9 yellow) had greater than 99% olive or yellow ancestry. We then ran a principal components analysis (PCA) of these 26 high coverage baboons with low admixture using PLINK [26] (—pca flag) using LD pruned markers (—indep-pairwise 50 5 0.1) to confirm if these selected animals clustered together (Fig 1A). The remaining high coverage animals that were not chosen to be part of this final reference panel were merged back into the query VCF using BCFTools [32]. We next ran a supervised analysis in ADMIXTURE [43] with K = 2 using the 26 high coverage, purebred individuals as a reference panel to obtain global ancestry estimates for each of the remaining target individuals.

The program RFMix [40] was used to estimate local ancestry using standard settings. For these analyses, RFMix [40] requires a balanced reference panel, so to create this we chose all 9 high coverage, purebred yellow individuals we had as they clearly formed their own cluster in the PCA (Fig 1A), and then selected 9 high coverage, purebred olives randomly from each of the represented olive clusters (Fig 1A) to capture all of the diversity within the colony. Karyotype plots were created using haptools v0.3.0 [49] after reformatting the RFMix [40] output files. RFMix [40] also outputs global ancestry estimates, which we used as comparison, and confirmation, to our ADMIXTURE [43] global ancestry results. Since RFMix [40] outputs

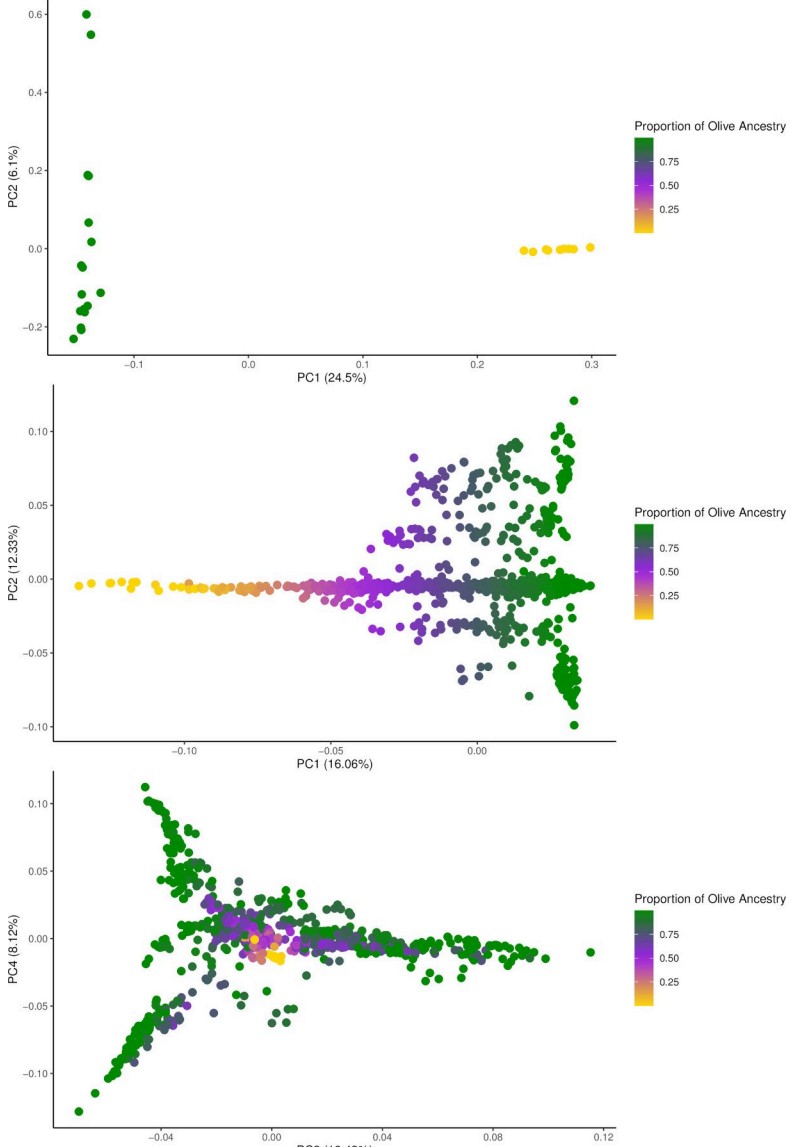

**Fig 1. Principal component analysis of the SNPRC baboon colony.** (A) PC1 and PC2 of the 26 purebred, founding olive or yellow animals identified using an unsupervised ADMIXTURE [43] run with K = 2 clustering. (B) PC1 and PC2 of the entire 881 colony sample set. (C) PC3 and PC4 of the entire 881 colony sample set. Plot colour based on the proportion of global olive ancestry. Figure generated in R v4.3.0 [47] using the ggplot2 package [48].

global ancestry per chromosome, we used Tractor [46] to extract the ancestry component of each sample and then used custom scripts to determine the global ancestry percentages.

In order to explore in more detail the effect of Tractor [46] correction in our entire sample, we compared the total number of local ancestry tracts observed after the first round of RFMix [40], unkinking with Tractor [46] using the original RFMix [40] output, a second round of RFMix [40] using the unkinked VCF files as input, and then finally, the unkinked data using Tractor [46] after a second round of local ancestry estimation using RFMix [40]. This is the recommended workflow presented in the Tractor paper resulting in the most accurate number of ancestry tracts based on simulated data [46]. We performed this analysis for both the Beagle [38, 39] and SHAPEIT5/IMPUTE5 [27, 41] pipelines.

## $F_{st}$ and AIMs

$F_{ST}$ values per marker were calculated using PLINK [26] (—fst and—within) using the 18 RFMix [40] reference animals. To create the AIMs panel, we selected markers with $F_{ST}$ values greater than 0.8 and binned them into 100,000 base pair windows across the autosomes, sorted them by value, and took the highest $F_{ST}$ value markers per window using a custom script. If there were multiple markers with identical $F_{ST}$ estimates within a window, we selected the first marker in the list. Because windowing can still theoretically lead to the inclusion of markers in LD, we sought to eliminate the potential of this occurring in two ways. First, we extracted this list of markers independently from the 9 olive and 9 yellow RFMix [40] reference animals highlighted above and ran strict LD pruning (—indep-pairwise 2 1 0.05) using PLINK [26]. Next, we merged the remaining marker set together and repeated the same LD pruning step, which resulted in a small set of markers (n = 1,747) spanning the entire genome that are ancestry informative. To confirm this, we ran a supervised ADMIXTURE [43] using the same 26 high-coverage, purebred, founding samples from the previous ADMIXTURE [43] step as references, using only the AIMs and compared these results to the initial supervised global ancestry analysis results obtained previously. We also wanted to compare the effectiveness of the AIMs in a wild population. We downloaded 27 wild baboon genomes published by Rogers et al. [2] (n = 6; 4 olives and 2 yellows) and Vilgalys et al. [20] (n = 21; 7 olives and 14 yellows) and ran ADMIXTURE [43] in unsupervised mode after LD pruning as above. Next, we used PLINK [26] to extract only the AIMs panel and reran an unsupervised global ancestry analysis using ADMIXTURE [43].

## Ethics statement

The data used in our study is publicly available data hosted on the NCBI Sequence Read Archive under BioProject PRJNA433868. No new biological samples were generated during this analysis. *In vivo* studies used to previously generate the genomic data were performed at the Southwest National Primate Research Center (SNPRC). The SNPRC is at the Texas Biomedical Research Institute (institutional assurance number D16-00048) and is an Association for Assessment and Accreditation of Laboratory Animal Care International (AAALAC) accredited program. All experiments were performed according to the provisions of the Animal Welfare Act of 1966 (Public Law 89–544), plus its subsequent amendments, as well as the standards set forth in the eighth edition of "The Guide for the Care and Use of Laboratory Animals". In addition, all research proposals dealing with animals must be approved by the Institutional Animal Care and Use Committee (IACUC) at Texas Biomedical Research Institute as required by the Health Research Extension Act of 1985 (Public Law 99–158). Verbal and written consent to analyze the data, along with a signed consensual data sharing agreement, was given to CK and BV by KS and SC on behalf of the SNPRC and Texas Biomedical Research Institute.

## Results

### Principal component analysis

We plotted the first two principal components of the 26 high coverage, purebred founders as this best represented the separation of olives and yellows in our dataset (Fig 1A). PC1 clearly differentiates the two baboon species and accounts for 24.5% of the variation seen within the data while PC2 captures variation within the olives and represents just 6.1% of the variation (Fig 1A). For the full dataset, admixed individuals occupy positions in the plot between purebred olive on one end and purebred yellow on the other across PC1 (Fig 1B). PC2 highlights

genetic diversity present in the olive baboons (Fig 1B). PC1 is responsible for 16.06% and PC2 12.33% of the variation seen in the dataset (Fig 1B). The two principal components combined illustrate a gradual shift in ancestry as the two ancestries converge into the middle representing roughly 50/50 hybrids (Fig 1B). PC3 (10.43%) and PC4 (8.12%) show limited diversity within the yellow samples and high levels of diversity exhibited by the olives, with hybrids largely occupying a similar space as the yellow baboons (Fig 1C).

## Imputation of low-coverage samples

After imputation and selection of high confidence imputed markers using the Beagle [38, 39] pipeline, there were 6,544,000 markers left for further analysis, a 17.34% improvement in the number of available markers (Fig 2) from the original low coverage dataset. The SHAPEIT5/ IMPUTE5 [27, 41] pipeline saw an even greater improvement in the number of recovered markers. A total of 6,616,489 were recovered, an 18.86% increase from the original low coverage marker set (Fig 2). VCF files for both pipelines after phasing. imputation, filtering, and phase correction are available for download at: https://zenodo.org/doi/10.5281/zenodo. 11176353.

## Global and local ancestry

Global ancestry estimates of the full baboon sample, using a supervised ADMIXTURE analysis [43] with the previously identified 26 unadmixed olive and yellow baboons as a reference sample were able to highlight errors in species assignment within the pedigree (Fig 3; S2 Table). Olive ancestry proportions estimated with ADMIXTURE [43] and RFMix [40] based on imputed data using the Beagle [38, 39] pipeline are highly correlated ($R^2 = 0.9189$), although, the estimates of olive ancestry obtained with RFMix [40] tend to be higher than those obtained with ADMIXTURE [43], particularly for individuals with low olive ancestry (S1 Fig). Comparing global ancestry estimates using RFMix [40] and ADMIXTURE [43] based on the SHA-PEIT5/IMPUTE5 [27, 41] pipeline has similarly strong correlation ($R^2 = 0.8876$) but is slightly lower than that observed with the Beagle [38, 39] pipeline (S2 Fig). We observe for this dataset that the global olive ancestry estimates are higher with RFMix [40] than ADMIXTURE [43] (S2 Fig). We also wanted to test if pipeline choice had an impact on the global ancestry estimates. We compared the outputs of ADMIXTURE [43] (S3 Fig) and RFMix [40] (S4 Fig) for both pipelines and found the correlations are in both cases excellent, with $R^2 = 0.9981$ and $R^2 = 0.9897$, respectively. Global ancestry estimates from ADMIXTURE [43] for each pipeline are provided in S2 Table for Beagle [38, 39] and S3 Table for SHAPEIT5/IMPUTE5 [27, 41].

Due to the unique nature of our dataset, which includes instances of hybridization in known pedigrees, we were interested in assessing local ancestry using RFMix [40]. We present a select group of sample karyograms depicting local ancestry tracts for a F1 hybrid (sample 10488) between putatively unadmixed parents (Fig 4A and 4B), as well as founders (samples 1X0110 and 1X1672) with little to no admixture (Fig 4C and 4D) based on global ancestry estimates. As expected, the F1 hybrid has chromosomes of both olive and yellow ancestry, whereas the unadmixed founders show chromosomes of singular ancestry with only minor contributions from the other species. It is important to note that there may be some misidentified ancestry blocks here for two reasons. First, RFMix [40] appears to overestimate the level of olive ancestry present compared to ADMIXTURE [43] (S1 and S2 Figs), and second, the reference panel used for this analysis was quite small due to the limited number of yellow animals in the colony. Also, there seem to be some phase switch errors in the F1 individual, reflected in switches of local ancestry in some of the chromosomes. This is something that we explore in more detail in the next section.

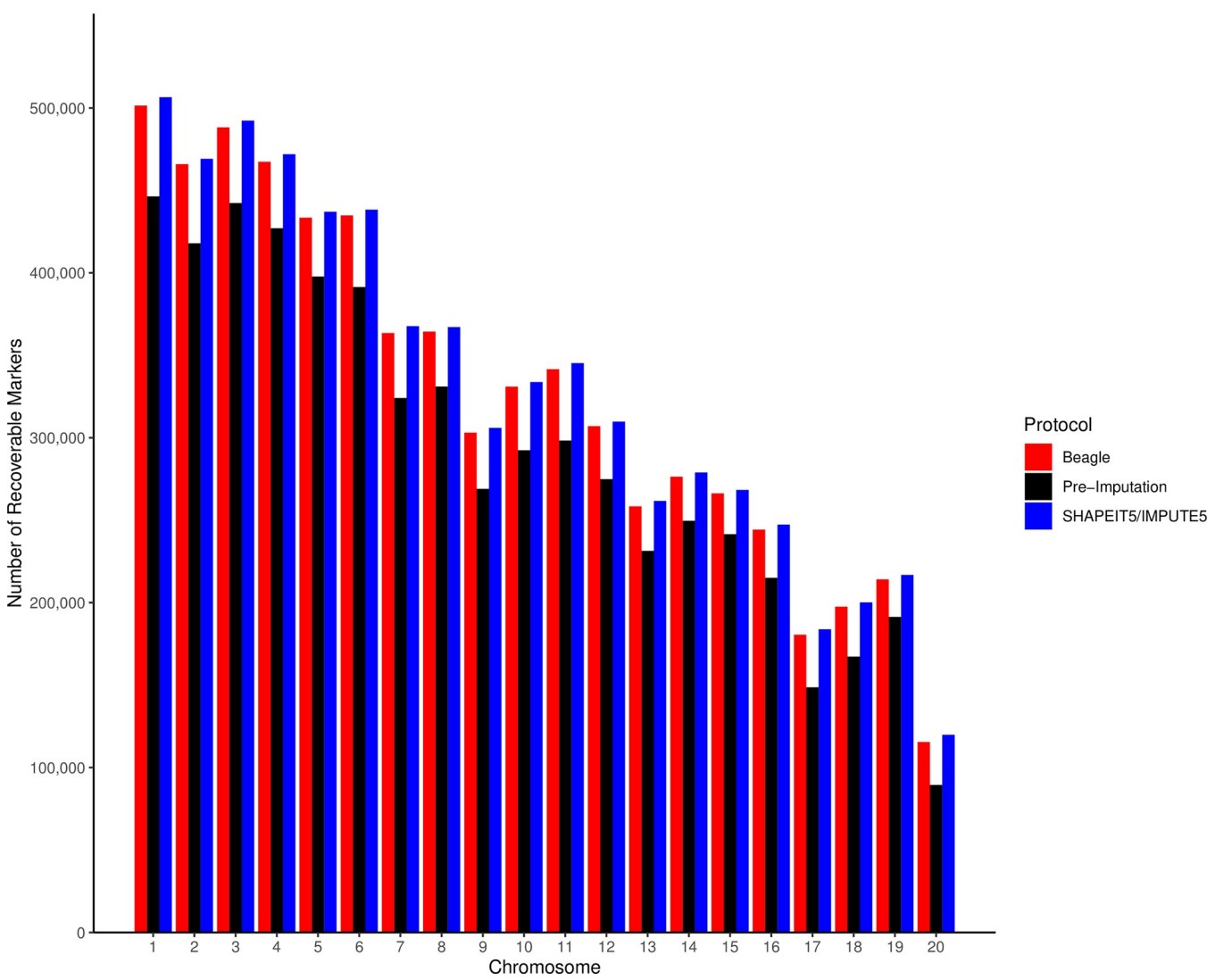

**Fig 2. Imputation of low coverage genotypes.** We ran two different imputation pipelines to improve the number of markers that pass quality control thresholds. The pre-imputation set is the marker baseline, while the Beagle [38, 39] and SHAPEIT5/IMPUTE5 [27, 41] protocols both substantially improve the number of markers for analysis, by 17.34% and 18.86%, respectively. Figure generated in R [47] using the ggplot2 package [48].

## Evaluation of phase switch errors based on local ancestry analyses

It has been reported that local ancestry information can be used to identify and correct for phase switch errors [46]. The availability of F1 crosses between unadmixed olive and yellow baboons makes it possible to evaluate potential phase switch errors. In theory, these individuals should have, for each chromosome pair, one chromosome of olive ancestry and the other of yellow ancestry. We observed evidence of phase switch errors, based on the local ancestry karyograms of an F1 hybrid (10488) with olive and yellow parents (Fig 4A). These errors are more evident with the Beagle [38, 39] pipeline (Fig 4A) than with the SHAPEIT5/IMPUTE5 [27, 41] pipeline (Fig 4B), likely because SHAPEIT5 [27] takes advantage of pedigree information for phasing. Notably, the use of successive cycles of local ancestry estimation using RFMix [40] combined with unkinking via Tractor [46] corrects many of these phase switch errors

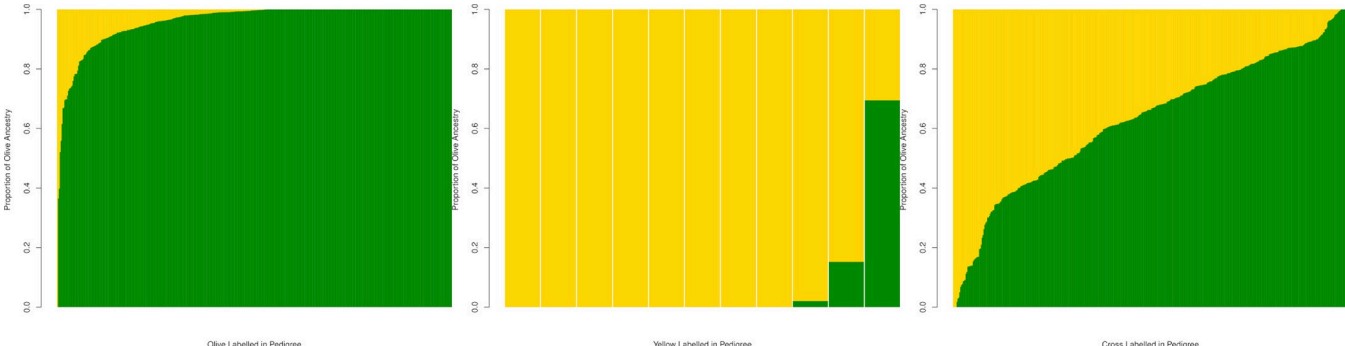

**Fig 3. Global ancestry estimates.** (A) Olive-labeled animals in the pedigree (n = 430), (B) yellow-labeled animals in the pedigree (n = 11), and (C) animals labeled as crosses in the pedigree (n = 440). Each putative ancestry label showcases at least one instance of incorrect labeling in the pedigree data. Barplots are sorted by percentage of estimated global olive ancestry as determined by ADMIXTURE [43] using supervised mode with K = 2 clusters. Figure generated in R [47].

(S6 Table), although based on the local ancestry karyograms of a parent-offspring trio (S6–S15 Figs), some errors do remain.

The results of the effects of Tractor [46] correction, in conjunction with repeated rounds of local ancestry estimation, are provided in S6 Table. Overall, after unkinking after the second round of RFMix [40], we observed a reduction in local ancestry switches of 34.95% in the

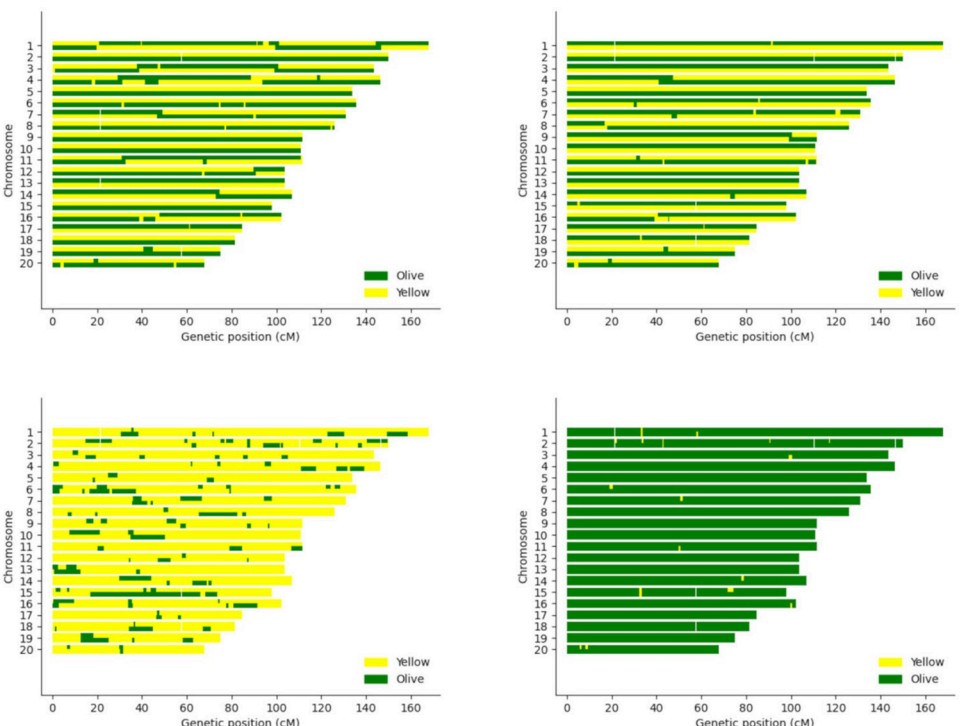

**Fig 4. Local ancestry estimates.** (A) Sample 10488, a first-generation hybrid of putative purebred olive and yellow parents. Local ancestry estimation was completed by RFMix [40] after running the Beagle [38, 39] pipeline. (B) 10488 after using the SHAPEIT5/IMPUTE5 [27, 41] pipeline. (C) Sample 1X0110, a putative purebred yellow founder with evidence of olive admixture. (D) A putative purebred olive founder, 1X1672. If not stated, the SHAPEIT5/IMPUTE5 pipeline data was used to generate the local ancestry karyograms. Green represents olive baboon ancestry and yellow represents yellow baboon ancestry. Karyograms generated using haptools [49].

Beagle [38, 39] and 22.59% in the SHAPEIT5/IMPUTE5 [27, 41] data, compared to the original RFMix [40] output. We also found that the number of local ancestry switches is systematically higher with the Beagle [38, 39] pipeline. After completing the full second local ancestry run followed by unkinking, the SHAPEIT5/IMPUTE5 [27, 41] pipeline has 24.51% less local ancestry switches than the Beagle [38, 39] pipeline. The local ancestry estimates for both rounds, including unkinking, are provided for both pipelines with the VCF files.

### $F_{st}$, AIMs, and genetic maps

Using the 9 purebred yellow and 9 purebred olive individuals from the RFMix [40] analysis, we estimated a weighted $F_{ST}$ of 0.397078 using the Weir and Cockerham method implemented in PLINK [26]. This value is slightly higher than what has been reported in prior studies of baboons from the SNPRC [19, 21]. A total of 27,940 markers are fixed ($F_{ST} = 1$) between the two species, an additional 3,940 more fixed markers than what has been previously reported [19]. The list of markers fixed between olive and yellow baboons and a genome-wide panel of 1,747 AIMs are also provided with the VCF files. We further compared the congruence between the global ancestry estimates obtained with the program ADMIXTURE [43] using the LD pruned full dataset and the AIMs panel (S4 Table). Both estimates show very high correlations, with an $R^2 = 0.9548$, although the AIMs panel tends to overestimate olive ancestry, especially in individuals with low olive ancestry, with respect to the genome-wide dataset (S5 Fig). S7 Table compares the output of the unsupervised ADMIXTURE [43] run for the whole LD pruned marker panel compared to just the AIMs from a sample of wild olive and yellow baboons from Africa [2, 20]. We report that even in wild baboon populations the AIMs do an excellent job of segregating ancestry in olive and yellow baboons, with $R^2 = 0.99723$ (S16 Fig; S7 Table). Lastly, we have provided genetic maps compatible with PLINK [26], SHAPEIT5 [27], and IMPUTE5 [41] using *Panubis1.0* coordinates with the VCF files.

## Discussion

In this paper, we present global and local ancestry estimates, as well as an $F_{ST}$ analysis of a large sample of olive and yellow baboons (n = 881) from the SNPRC, for which whole genome data are available. This sample represents a challenging dataset to work with because of founder effects associated with the creation of the baboon colony, as well as admixture and inbreeding experiments leading to elevated levels of relatedness between the individuals that would likely not be representative of a true wild population. An additional complication is related to the substantial differences in sequencing coverage for the individuals of the sample, with most of the individuals sequenced at relatively low coverage.

In this respect, previous studies have highlighted the benefits of doing imputation of low-coverage samples using high-coverage sequences as a reference sample [44, 50–52]. The advantage of such an approach is that missing variants in the low-coverage sample sets can be obtained with high accuracy post-imputation, improving both the quantity and quality of markers available for analysis. Imputation is a particularly attractive strategy in the SNPRC sample because there is high coverage sequencing data for all labeled founders in our dataset, with minimum depth of 29X (S1 Table). By doing imputation, we were able to obtain high quality genotype calls for common genetic markers with no missing data. We opted to initially do genotype refinement, phasing, and imputation using Beagle [38, 39] because of its reported accuracy [44, 45] and supplemented this with an additional phasing and imputation pipeline using SHAPEIT5 [27] and IMPUTE5 [41] to test improvement in phasing accuracy using pedigree information.

The availability of local ancestry estimates from RFMix [40] made it possible to evaluate potential phase switch errors. In theory, F1 crosses between unadmixed olive and yellow baboons should have, for each chromosome pair, one chromosome of olive ancestry and the other of yellow ancestry. We observed evidence of phase switch errors in our dataset, which tend to be more pronounced when using the Beagle [38, 39] (Fig 4A; S8 Fig) rather than the SHAPEIT5/IMPUTE5 [27, 41] (Fig 4B; S12 Fig; S6 Table) pipeline, presumably because the latter can use pedigree information for phasing. Strategies based on local ancestry have been proposed to correct these, and we found that many of the putative phasing errors are corrected using Tractor [46] (S11 and S15 Figs; S6 Table). Of note, for both pipelines, in a few regions of the genome there are segments of only olive or yellow ancestry, instead of the expected dual olive and yellow segments in sample F1 hybrid genomes (Fig 4A and 4B; S11 and S15 Figs). This may be due to some limited admixture in the parental baboons (Fig 4C and 4D; S6 and S7 Figs; S2, S3 and S5 Tables), or alternatively, possible errors in local ancestry estimation (S5 Table).

Previous studies have shown evidence of gene flow between olive and yellow baboons in the wild. Field research has documented ongoing hybridization in wild populations at several national parks between many different baboon species [53–59]. Recent genomic studies also support these observations of hybridization. Evidence of gene flow between olive and yellow baboons has previously been reported at Amboseli [21] and a recent study [20] reported the presence of olive admixture in the yellow founders used to begin the breeding program at the SNPRC. Further, across 19 different regions in Africa, wild baboons frequently exhibited evidence of admixture [3]. We showcase potential evidence of this in an F1 hybrid individual with putatively unadmixed parents (Fig 4A and 4B, sample 10488), and in SNPRC-labeled purebred, founding yellows (Fig 4C; S6 Fig, samples 1X0110 and 1X0102). These examples suggest that some of the hypothetically purebred founders had some ancestry from other species. However, we cannot eliminate the possibility of local ancestry errors, particularly considering that the number of reference samples used to estimate local ancestry with the program RFMix [40] was quite small, due to the very limited number of yellow founders, and that the olive baboon reference samples showed higher genetic diversity than the yellow baboon samples (Fig 1A). In this respect, it is important to note that the RFMix [40] global olive ancestry estimates for both pipelines tend to be higher than those observed with the program ADMIXTURE [43], although they show very strong correlations with one another (S1 and S2 Figs).

To identify putative local ancestry errors, we analyzed local ancestry inconsistencies of a trio comprising an F1 cross (1X3837) between two relatively unadmixed olive (1X0026) and yellow (1X0102) founder baboons (S6–S15 Figs; S5 Table). More specifically, we explored the presence of Mendelian incompatibilities of the F1 offspring and parental local ancestry estimates where the F1 has two copies of either olive or yellow ancestry while the olive and yellow parent both have two copies of their respective ancestries. In this situation, the F1 is expected to have one copy of each ancestry in this region. Evaluating this for all 20 chromosomes, we observed that 3.55% of the markers show local ancestry inconsistencies (S5 Table), which is a relatively small proportion of the whole genome. When we do see these inconsistencies, they tend to be in ancestry tracts with a small number of markers (S5 Table). It should be noted that there are some chromosomal regions where the F1 individual (both with and without Tractor [46] correction) shows olive ancestry in both chromosomes and the parental yellow also shows instances of olive ancestry. This would suggest the presence of a small proportion of olive ancestry in this yellow founder, in agreement with what has been reported in other studies of SNPRC yellow founders [20]. In summary, our analyses show evidence of both local ancestry errors of relatively small magnitude and limited olive ancestry in some of the yellow founders.

We carried out detailed global ancestry analyses using both ADMIXTURE [43] and RFMix [40] and observed very consistent results with $R^2$ values greater than 0.88 (S1 and S2 Figs) between the two software packages. Running two different phasing/imputation pipelines allowed us to compare the results for both, and we observed that the global ancestry estimates were almost identical using ADMIXTURE [43] and RFMix [40] (S1 and S2 Figs). Clearly, when comparing the global ancestry results obtained with the two software packages and the two phasing/imputation pipelines, the effect of the global ancestry software is considerably stronger than the effect of the phasing/imputation pipelines (S3 and S4 Figs). The Beagle [38, 39] pipeline had overall higher agreement between the two global ancestry estimation programs (S1 Fig) but at the cost of a slightly lower number of recovered markers after imputation (Fig 2) and more evidence of phase switch errors in our local ancestry results (Fig 4A; S8 Fig; S6 Table). In contrast, the SHAPEIT5/IMPUTE5 [27, 41] protocol recovered more markers (Fig 2) and had fewer phase switch errors (Fig 4B; S12 Fig; S6 Table); however, it did not perform as well when comparing global ancestry estimation between ADMIXTURE [43] and RFMix [40] (S2 Fig).

There are 101 instances where the pedigree information shows some discrepancies with the global ancestry estimates (S2 Table). For example, the sample 1X4384, labeled as an olive in the pedigree, was identified as having purebred yellow ancestry in our analyses, similarly to what has been reported in a previous study [19]. There are 50 instances where olive-labeled animals have olive ancestry less than 90%, and two yellow samples have less than 90% yellow ancestry (Fig 3A and 3B; S2 Table). When examining the crosses, we identified 14 samples with yellow ancestry estimates greater than 90%, four with over 99% yellow ancestry, and 35 samples with olive ancestry greater than 90%, 11 of those with more than 99% olive ancestry (Fig 3C; S2 Table). Lastly, as highlighted previously, only 26 of the founding animals were purebred according to our unsupervised ADMIXTURE [43] run, despite all of the olive or yellow founders being labeled as putatively purebred in the pedigree (S2 Table). This may possibly reflect hybridization of baboon species historically in the wild, although it is important to keep in mind that there are inherent errors in the estimation of global and local ancestry proportions.

Our unsupervised global ancestry estimates from ADMIXTURE [43] identified 9 yellow founders with very high yellow ancestry (Fig 1A; S2 Table). Of note, an earlier report by Robinson et al. [19] also used an unsupervised global ancestry method on many of the same founder samples that are in our study and they too documented little to no admixture present in the founders of the SNPRC. However, our local ancestry analyses indicate that, in agreement with previous analyses [20, 21], some of the yellow founders may have olive ancestry as a result of hybridization in the wild (Fig 4C; S6 Fig; S5 Table). As stated previously, the global ancestry analyses based on local ancestry results generated with RFMix [40] tend to be higher than the global ancestry estimates obtained with ADMIXTURE [43], so we cannot eliminate the possibility that the RFMix [40] results are biased towards higher estimates of olive ancestry (S1 and S2 Figs), perhaps due to the small reference samples used in the local ancestry analyses and the larger genetic diversity observed in the olive founders included in the reference panel (Fig 1A).

Olive and yellow baboons show very high genetic differentiation, and we observed a weighted $F_{ST}$ value of 0.397078 between unadmixed yellow and olive founders. We found that over 27,000 markers have alleles that are fixed in each species. Additionally, we developed a genome-wide panel of AIMs including more than 1,700 genetic markers, that can be used to reliably estimate olive and yellow ancestry and are very strongly correlated with a considerably larger genome-wide panel of SNPs in both captive and wild populations (S5 and S16 Figs; S4 and S7 Tables). However, we noticed that in individuals with low olive ancestry, the olive ancestry estimates based on the AIMs panel are slightly higher than those produced with the denser genome-wide panel (S5 Fig).

## Conclusions and limitations

To summarize, we provide here a large collection of baboon genomic data. The animals from the SNPRC are an ideal resource for those interested in obtaining biomedical, primatological, hybridization, or inbreeding data. Genomic research on baboons has seen an increase in interest over the last few years, presumably due to their similarities to modern humans in many regards. We have genetically cataloged 881 baboons from the SNPRC to high resolution, provided global ancestry estimates for each sample and provided local ancestry estimates for 863 individuals in our dataset. We updated resources for future use for anyone interested in studying these two baboon species, including new genetic maps with *Panubis1.0* coordinates in multiple formats, a list of more than 27,000 fixed markers between olive and yellow baboons, and a concise set of 1,747 AIMs that are highly diagnostic between the two species. The availability of genotype data, in combination with global and local ancestry information, will facilitate different types of statistical analyses, including admixture mapping, association studies conditioning on local ancestry, or joint ancestry and association tests.

Our study has several limitations. A major limitation is the relatively small sample size, which has several implications for our analyses. First, having a small sample negatively affects the accuracy of statistical phasing, and we do see evidence of phase switch errors when evaluating ancestry switches in F1 hybrids (S8–S15 Figs). We show that a pedigree-aware protocol using SHAPEIT5 [27] and IMPUTE5 [41] resulted in phasing improvements with respect to the Beagle [38, 39] pipeline (Fig 4A and 4B; S8 and S12 Figs; S6 Table). Also, in agreement with previous reports [46], we observe that it is possible to leverage local ancestry information to partially correct instances of phase switch errors (S11 and S15 Figs; S6 Table). Second, the very small number of yellow unadmixed individuals in the SNPRC pedigree (n = 9), which constrained the size of the reference panels used to estimate local ancestry, may have influenced the accuracy of the local ancestry estimates A detailed analysis of local ancestry inconsistencies in a trio confirms the presence of local ancestry errors (S5 Table), however, this represents less than 4% of the total number of markers. Utilizing a larger reference panel, possibly including a more diverse sample from outside the SNPRC, may help improve some key steps of our analyses, such as the accuracy of our local ancestry estimations, and clarify the extent of admixture in the SNPRC olive and yellow founders more effectively. As discussed above, there may be olive ancestry in some of the yellow SNPRC samples because of gene flow between both species in the wild (Figs 3B and 4C; S6 Fig), which was highlighted in previous studies. Another limitation of our study is the substantial difference in sequencing coverage of the SNPRC samples. We tackled this limitation by performing imputation of the low-coverage samples using high-coverage genomes, which allowed us to build a high-quality dataset including over 6.5 million common markers without missing data. Overall, we illustrate high correlations between the global ancestry estimates obtained with ADMIXTURE [43] and RFMix [40] after phasing and imputation using Beagle [38, 39] (S1 Fig) or SHAPEIT5/IMPUTE5 [27, 41] (S2 Fig). However, there is evidence of phase switch errors in our data even after implementing Tractor [46] to correct these errors (Fig 4A and 4B; S11 and S15 Figs; S6 Table), although the amount of these errors is still relatively small comparatively to the whole genome. Despite these limitations, it is our hope that the materials and resources provided here will assist future researchers interested in studying baboons genomically.

## Supporting information

**S1 Fig. Beagle pipeline RFMix vs. ADMIXTURE global ancestry comparison.** Global olive ancestry estimates from both ADMIXTURE [43] and RFMix [40] using the Beagle [38, 39] pipeline. There is excellent agreement between the two software with $R^2$ = 0.9189. RFMix [40]

appears to overestimate olive ancestry relative to ADMIXTURE [43]. Plot was generated in PAST v4.03 [60].
(TIF)

**S2 Fig. SHAPEIT5/IMPUTE5 pipeline RFMix vs. ADMIXTURE global ancestry comparison.** Global olive ancestry estimates from both ADMIXTURE [43] and RFMix [40] using the SHAPEIT5/IMPUTE5 [27, 41] pipeline. There is a slight drop in the congruence between the ancestry estimates compared to the Beagle pipeline with $R^2 = 0.8876$. As seen in S1 Fig, RFMix [40] overestimates olive ancestry compared to ADMIXTURE [43]. Plot was generated using PAST [60].
(TIF)

**S3 Fig. Comparison of the ADMIXTURE global ancestry estimates from both pipelines.** Global olive ancestry estimates using ADMIXTURE [43] from both pipelines. The software has an extremely strong association with $R^2 = 0.9982$. Plot was generated in PAST [60].
(TIF)

**S4 Fig. Comparison of the RFMix global ancestry estimates from both pipelines.** Global olive ancestry estimates using RFMix [40] from both pipelines. The software has a strong association with $R^2 = 0.9897$ but shows less consistency in ancestry assignment than ADMIXTURE [43] when comparing both pipelines (S3 Fig). Plot was generated using PAST [60].
(TIF)

**S5 Fig. Comparison of the ADMIXTURE global ancestry estimates using the full dataset compared to the AIMs-only dataset.** Global olive ancestry estimates from ADMIXTURE [43] using the Beagle pipeline [38, 39] LD pruned full marker set (n = 354,064) vs. just the AIMs (n = 1,747). The association is very strong ($R^2 = 0.9548$), however, the AIMs overestimate olive ancestry, especially at low levels of ancestry compared to the full marker set. Plot was generated in PAST [60].
(TIF)

**S6 Fig. Local ancestry karyogram for 1X0102.** Sample 1X0102 is a purebred yellow founder (S2 Table). His local ancestry estimation based on RFMix [40] that has been corrected for phase switching using Tractor [46] showcases some historic olive ancestry. Plot was generated using haptools [49].
(TIF)

**S7 Fig. Local ancestry karyogram for 1X0026.** Sample 1X0026 is a purebred olive founder (S2 Table). Her local ancestry estimation that was estimated with RFMix [40] and corrected for phase switches using Tractor [46] largely agrees with the ADMIXTURE [43] global ancestry estimates. Plot was generated using haptools [49].
(TIF)

**S8 Fig. Original Beagle local ancestry karyogram for 1X3837.** Sample 1X3837 is a first-generation hybrid offspring of a purebred olive (1X0026) mother and purebred yellow (1X0102) father. This plot, generated by haptools [49] shows the original output by Beagle [38, 39].
(TIF)

**S9 Fig. Unkinked Beagle local ancestry karyogram for 1X3837.** The same original Beagle [38, 39] data as S8 Fig but unkinked using Tractor [46]. Plot generated by haptools [49].
(TIF)

**S10 Fig. Beagle local ancestry karyogram for 1X3837 after two local ancestry runs.** Beagle [38, 39] data with a second RFMix [40] round using the unkinked data from S9 Fig as input.

The plot was generated using haptools [49].
(TIF)

**S11 Fig. Beagle local ancestry karyogram for 1X3837 after two local ancestry runs and unkinking.** Beagle [38, 39] data with a second RFMix [40] round using the unkinked data from S9 as input followed by a second round of unkinking using Tractor [46]. Her local ancestry estimation done by RFMix [40] and corrected for phase switch errors using Tractor [46] still shows evidence of switch errors. Additionally, many of these switches do not align with what is seen in the parental karyograms (S6 and S7 Figs). Plot created using haptools [49].
(TIF)

**S12 Fig. Original SHAPEIT5/IMPUTE5 local ancestry karyogram for 1X3837.** This plot, generated by haptools [49], shows the original output by SHAPEIT5/IMPUTE5 [27, 41].
(TIF)

**S13 Fig. Unkinked SHAPEIT5/IMPUTE5 local ancestry karyogram for 1X3837.** The same original SHAPEIT5/IMPUTE5 [27, 41] data as S12 Fig but unkinked using Tractor [46]. Plot generated by haptools [49].
(TIF)

**S14 Fig. SHAPEIT5/IMPUTE5 local ancestry karyogram for 1X3837 after two local ancestry runs.** SHAPEIT5/IMPUTE5 [27, 41] data with a second RFMix [40] round using the unkinked data from S13 Fig as input. The plot was generated using haptools [49].
(TIF)

**S15 Fig. SHAPEIT5/IMPUTE5 local ancestry karyogram for 1X3837 after two local ancestry runs and unkinking.** SHAPEIT5/IMPUTE5 [27, 41] data with a second RFMix [40] round using the unkinked data from S13 Fig as input followed by a second round of unkinking using Tractor [46]. While some switch errors and ancestry misassignments are still persistent, these are reduced relative to S11 Fig. Plot created using haptools [49].
(TIF)

**S16 Fig. Comparison of the ADMIXTURE global ancestry estimates using the full dataset compared to the AIMs-only dataset of wild baboon populations.** Global olive ancestry estimates from ADMIXTURE [43] using an LD pruned marker panel (n = 334,469 for Vilgalys et al. [20] samples; n = 152,878 for Rogers et al. [2] samples) vs. just our panel of AIMs (n = 1,747). The association is extremely strong ($R^2$ = 0.99723). The AIMs still tend to overestimate the amount of olive ancestry even in these wild populations. Plot was generated in PAST [60].
(TIF)

**S1 Table. Mean depth per sample.** Data derived from VCFtools [42] using the—depth flag.
(XLSX)

**S2 Table. ADMIXTURE global ancestry estimates from the Beagle pipeline merged with the pedigree data.** Pedigree information includes generation of each sample and the SNPRC-labeled taxon. Reference panel animals (n = 51) are bolded while red highlighted rows (n = 101) are samples with unexpected ancestry given their pedigree label. Samples used as the RFMix [40] reference panel (n = 18) are noted in column F.
(XLSX)

**S3 Table. ADMIXTURE global ancestry estimates from the SHAPEIT5/IMPUTE5 pipeline.**
(XLSX)

**S4 Table. ADMIXTURE global ancestry estimates using the Beagle pipeline full data compared to the Beagle pipeline AIMs-only data.**
(XLSX)

**S5 Table. Local ancestry calls generated by RFMix for one parent-offspring trio illustrating local ancestry errors.** Local ancestry calls generated using RFMix [40] for 3 baboons from the SNPRC using the SHAPEIT5/IMPUTE5 pipeline [27, 41] with phase switch error correction using Tractor [46]. Yellow highlights showcase Mendelian ancestry misassignments by RFMix [40] based on parental ancestry blocks. Data has had two runs of RFMix [40] and Tractor [46] correction like the data in S15 Fig.
(XLSX)

**S6 Table. Ancestry block statistics.** Statistics for several rounds of RFMix [40] local ancestry estimation and Tractor [46] unkinking for both phasing/imputation pipelines utilizing full local ancestry dataset (n = 863).
(XLSX)

**S7 Table. ADMIXTURE global ancestry estimates comparing wild olive and yellow baboon whole genome ancestry estimates against AIMs-only ancestry estimates.**
(XLSX)

**S1 File.**
(XLSX)

## Acknowledgments

We would like to thank Jenny Tung for providing detailed information for the animals sequenced in [20] that we used as a test of our AIMs in wild populations.

## Author Contributions

**Conceptualization:** Christopher Kendall, Esteban Parra, Michael Schillaci, Bence Viola.

**Data curation:** Christopher Kendall.

**Formal analysis:** Christopher Kendall, Jacqueline Robinson, Guilherme Debortoli.

**Funding acquisition:** Christopher Kendall, Esteban Parra, Michael Schillaci, Bence Viola.

**Investigation:** Christopher Kendall, Debbie Christian, Deborah Newman, Kenneth Sayers, Shelley Cole.

**Methodology:** Christopher Kendall, Jacqueline Robinson, Esteban Parra, Michael Schillaci, Bence Viola.

**Project administration:** Christopher Kendall, Esteban Parra, Michael Schillaci, Bence Viola.

**Resources:** Christopher Kendall, Jacqueline Robinson, Debbie Christian, Deborah Newman, Kenneth Sayers, Shelley Cole.

**Software:** Christopher Kendall, Guilherme Debortoli, Amin Nooranikhojasteh.

**Supervision:** Esteban Parra, Michael Schillaci, Bence Viola.

**Validation:** Christopher Kendall.

**Visualization:** Christopher Kendall.

**Writing – original draft:** Christopher Kendall, Esteban Parra, Michael Schillaci, Bence Viola.

**Writing – review & editing:** Christopher Kendall, Debbie Christian, Deborah Newman, Kenneth Sayers, Shelley Cole, Esteban Parra, Michael Schillaci, Bence Viola.

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
