## [Decision Letter · Decision Letter 0]

11 Apr 2024

PONE-D-24-01461Global and Local Ancestry Estimation in a Captive Baboon ColonyPLOS ONE

Dear Dr. Kendall,

Thank you for submitting your manuscript to PLOS ONE. After careful consideration, we feel that it has merit but does not fully meet PLOS ONE’s publication criteria as it currently stands. Therefore, we invite you to submit a revised version of the manuscript that addresses the points raised during the review process.

**As you can see that we managed to get only one review of the manuscript, however, we think tha it is quite detailed and indepth review. To save the time, we request you to revise the manuscript by considering the suggestions of the reviewer.**

We look forward to receiving your revised manuscript.

Kind regards,

Gyaneshwer Chaubey

Academic Editor

PLOS ONE

Journal Requirements:

CK was supported by an NSERC CGS-D grant (CGSD2 - 535025 - 2019) for the duration of this research (https://www.nserc-crsng.gc.ca/index_eng.asp). The establishment, maintenance, and biological characterization of the pedigreed baboon colonies at the Southwest National Primate Research Center of Texas Biomedical Research Institute (SNPRC at Texas Biomed) has been supported in large part by grants to Texas Biomed Investigators by the National Institutes of Health (P51 RR013986, P01 HL028972) (https://www.nih.gov/) (DC, DN, KS, SC).

Reviewers' comments:

Reviewer's Responses to Questions

**Comments to the Author**

1. Is the manuscript technically sound, and do the data support the conclusions?

Reviewer #1: Yes

2. Has the statistical analysis been performed appropriately and rigorously? 

Reviewer #1: N/A

3. Have the authors made all data underlying the findings in their manuscript fully available?

Reviewer #1: Yes

4. Is the manuscript presented in an intelligible fashion and written in standard English?

Reviewer #1: Yes

5. Review Comments to the Author

**Reviewer #1:** The authors propose analyzing large-scale genomic data from a captive population of baboons, using genomic data from 881 specimens, to build a model for studying ancestry, considering the hybridization of individuals of the Olive and Yellow species. The groups are unbalanced, with a reduced number of yellow species.

Several software programs are used for this, as well as filtering strategies and the selection of representatives for Phasing and Imputation. Although the methodology is coherent, it isn't easy to follow the line of reasoning to fully understand the motivation and the sequence of steps to reach the final goal. Many analyses are repeated with more than one approach, which, on the one hand, confirms each other and, on the other, makes it difficult to read and understand the study. The successive dismemberment of the set in order to select individuals from the original population could also be better structured.

Two sets of markers are defined and made available in various formats - a large set with 27,000 markers and a smaller one with 1,747 measurements with high discriminatory power to assess the degree of ancestry concerning the involved species (O/Y).

I believe this work should contribute to future studies. The authors have structured the data and analyzed several generations of crosses between the species. However, questions remain, some of which the authors identify in the limitations and others which stay open and refer to the present study. Here are some of them.

1) What is the founders' composition regarding olive and yellow baboons' quantity? Is it necessary to cluster them, considering they are already labeled (pedigree)?

2) Aren't some common sequence regions expected to be in both species? Have the authors considered some other phylogenetic information involving the species? Which is regarded as the more basal, for instance? Or how close they are (relative to other baboon species)?

3) Can choosing a more specific cluster—as was done in the study—create a bias absent in wild conditions? So, can this step be justified?

4) The above questions also apply to possible corrections of phasing errors. Could phasing errors be confused with actual ancestral genetic information?

6. PLOS authors have the option to publish the peer review history of their article (what does this mean?). If published, this will include your full peer review and any attached files.

Reviewer #1: No

---

## [Author Response · Author response to Decision Letter 0]

10 May 2024

We would like to thank PLoS One, Gyaneshwer Chaubey our academic editor, and the anonymous reviewer for providing such thorough feedback on our submission. We were delighted to see that we only had minor revisions to make to our manuscript. Below you will find our responses to each of the concerns the academic editor and the reviewer raised about our manuscript. 

The academic editor first pointed out that some of our file names and some formatting did not adhere to PLoS One’s formatting. We have edited our manuscript, which can be found in the track changes document and the updated manuscript, which reflect these changes. We have also re-uploaded our files with the correct naming conventions as described in the style templates. Next, they required us to amend our funding statement for all internal and external funding for each author that was relevant to the study. This has been updated in our funding statement and attached to the cover letter as requested. The editor asked us to confirm our stance on our data availability and there has been no change, our data is publicly available upon acceptance at https://zenodo.org/doi/10.5281/zenodo.10493366. Lastly, the academic editor asked us to include an ethics statement and to ensure our reference list and citations were correct. We have added the ethics statement to the ‘Methods’ section as requested, and the manuscript citations and references have been checked for accuracy. Any changes here are reflected in the track changes document.

Below, we have included answers to the reviewer’s comments done in turn as they were given to us. The reviewer’s comments are starred, and our answers are in normal type below each comment.

**The authors propose analyzing large-scale genomic data from a captive population of baboons, using genomic data from 881 specimens, to build a model for studying ancestry, considering the hybridization of individuals of the Olive and Yellow species. The groups are unbalanced, with a reduced number of yellow species. Several software programs are used for this, as well as filtering strategies and the selection of representatives for Phasing and Imputation. Although the methodology is coherent, it isn't easy to follow the line of reasoning to fully understand the motivation and the sequence of steps to reach the final goal. Many analyses are repeated with more than one approach, which, on the one hand, confirms each other and, on the other, makes it difficult to read and understand the study. The successive dismemberment of the set in order to select individuals from the original population could also be better structured.**

We thank the reviewer for their comments. We are pleased that they find the methodology is coherent. Upon further review, we agree with the reviewer that this section can be a bit difficult to follow due to the repeated changes in methodology between the two datasets. We have made edits in the manuscript to make this clearer. This can be seen in lines 194-199 where justification for relaxing of the minor allele frequency (MAF) filter was made. Lines 212-220 have added justification for the second phasing and imputation step. Additionally, lines 393-414 (and figures referenced therein) describe our results in detail of why the second phasing pipeline was needed due to phase switching. Lines 248-265 discussing RFMix have been updated. The separation of the two analysis sets (the high vs. low coverage panels) have been reworked considerably at lines 178-211 to explain any differences in methodology.

**Two sets of markers are defined and made available in various formats - a large set with 27,000 markers and a smaller one with 1,747 measurements with high discriminatory power to assess the degree of ancestry concerning the involved species (O/Y). I believe this work should contribute to future studies. The authors have structured the data and analyzed several generations of crosses between the species. However, questions remain, some of which the authors identify in the limitations and others which stay open and refer to the present study. Here are some of them.**

**1) What is the founders' composition regarding olive and yellow baboons' quantity? Is it necessary to cluster them, considering they are already labeled (pedigree)?**

The founders’ composition includes 45 labelled olives and 11 labelled yellows, of which we used 9 yellows and 42 olives as our founders for our study due to them being part of the high coverage panel (>= 15X depth). We have edited this in our manuscript, so it is clear for readers at the onset on line 184. We feel the clustering is necessary for several reasons. First, it confirms that we only have two genetic ancestries in the founding populations. Secondly, as was highlighted in Robinson et al. [1], and later confirmed in our paper, it was not uncommon for the pedigree labels to be inconsistent. For this reason, we believe that the initial clustering step was important for further downstream analyses such as supervised global and local ancestry estimates. As shown in S2 Table, there are many instances of founding animals with pedigree labels that do not match what is seen in the global ancestry estimation. Relying solely on pedigree labels for later supervised analyses would have led to skewed ancestry definitions of the query panels. 

**2) Aren't some common sequence regions expected to be in both species? Have the authors considered some other phylogenetic information involving the species? Which is regarded as the more basal, for instance? Or how close they are (relative to other baboon species)?**

The two baboon species in our study show diverged evolutionary histories, with SNV data illustrating a north-south clade division at approximately 1.4 million years ago, with P. anubis occupying the north and P. cynocephalus occupying the south, as illustrated in Rogers et al. [2]. Additional whole genome and mitochondrial data supports this, which is discussed in Sørensen et al. [3]. Older evidence from Zinner et al. [4] shows cytochrome b divergences between Papio spp. to a northern and southern clade ~2.1 million years ago. These relatively deep inferred split times are consistent with the high genetic differentiation observed between both species. As reported in the manuscript, the FST values we estimated based on genome-wide data are quite high (FST ~ 0.4), which is consistent with prior estimates. We have added the divergence information to the ‘Introduction’ section of the manuscript on lines 50-58 but feel that exploration of the phylogenetic information is outside the scope of this paper considering we focused solely on olive and yellow baboons. A proper study of this would require samples from all the extant species, for which we do not have data for. 

**3) Can choosing a more specific cluster—as was done in the study—create a bias absent in wild conditions? So, can this step be justified?**

As described in the paper, the selection of AIMs was based on FST values, which were estimated between 9 purebred olive and 9 purebred yellow baboons. Importantly, consistent with the high genetic differentiation between olive and yellow baboons, almost 28,000 markers show an FST equal to 1 (fixed) between both species, which means that all olive and yellow baboons have different alleles at these sites. When preparing the final list of AIMs for admixture inference and admixture mapping studies covering the entire genome, we followed an approach that ensures that first, all the markers in the AIM panel are independent in both species with no linkage disequilibrium between any pair of markers between them. Secondarily, that all the markers show extremely high differentiation between both species with FST >= 0.8. In theory, this AIMs panel should perform well also in wild yellow and olive baboons, and yellow/olive hybrid zones. We tested the AIMs panel against two independent sets of wild baboon data (previously published data from [2,5]) and found that the AIMs have excellent predictive power in classifying ancestry between olive and yellow baboons even in wild populations, despite being trained on captive populations. Lines 283-288 explain the methodology for this section while lines 429-433 and 548-552 describe the results. Additional Supporting Information in the form of S7 Table, which contains the ADMIXTURE global ancestry estimates from the wild populations using whole genome LD pruned markers compared to our AIMs panel, and Fig S16 which shows the linear regression of the same data have been created. The AIMs in our captive dataset have an R2 value greater than 0.95 compared to the whole-genome panel [S5 Fig] and R2 = 0.99723 in the wild populations [S16 Fig]. 

**4) The above questions also apply to possible corrections of phasing errors. Could phasing errors be confused with actual ancestral genetic information?**

The reviewer raises a very important point regarding the relative role of phasing errors vs. errors in the estimation of local ancestry, which is highly relevant for this type of study. There have been numerous studies exploring Switch Error Rates (SER) in statistical phasing, which can have substantial consequences in downstream analyses [6-8]. These error rates depend on different factors, primarily sample size and allele frequency [6-8]. In this respect, it is important to note that our sample is relatively small with less than 1,000 individuals, so phase switch errors are expected. However, it has been reported that it is possible to use local ancestry information to correct these errors, and we have used the method implemented in Tractor [9] to correct phase switch errors in our sample. Additionally, the accuracy of local ancestry estimates depends on phasing accuracy (although RFMix seems to be quite robust in this regard) and the size and composition of the reference sample [10]. As we describe in the paper, our reference sample (9 yellow and 9 olive baboons) is relatively small due to the limited number of yellow baboon founders, and we highlight this as one of the major limitations of our study. Unfortunately, it is not possible for us to know with precision the SER or local ancestry errors in this baboon sample. 

Despite this, analyzing a first-generation (F1) cross between two relatively unadmixed olive (~100% olive) and yellow (~90% yellow) founder baboons (estimates based on RFMix [10] local ancestry rather than global ancestry), we can get some useful information about the extent of phase switch errors and local ancestry inconsistencies. In theory, an F1 cross between unadmixed olive and yellow baboons should show one chromosome of each ancestry. However, in our case, because the yellow founder shows some evidence of olive ancestry, we would expect a few regions where the two chromosomes have olive ancestry. If there is evidence of ancestry switches in the F1 individual that are absent in the parents, these can be primarily attributed to phase switch errors. Also, we can track the proportion of markers where the local ancestry of the F1 individual is inconsistent with the local ancestry of the parents, based on Mendelian inheritance. After performing these analyses, it is evident that both phase switch errors and local ancestry inconsistencies are observed in our sample, although it does not seem that these are substantial. 

In the new figures (S6-S15 Figs), we show the local ancestry karyograms of the parental olive baboon (~100% olive, S7 Fig), parental yellow (~90% yellow, S6 Fig), and the F1 offspring (S8-S15 Figs). For the F1 individual, we include karyograms for different rounds of RFMix [10] and phase correction (i.e. unkinking) using Tractor [9]. We would like to note that these local ancestry estimations are different from the global ancestry estimates we provide in S2 Table, which can mostly be explained by differences in the software calculations. Without Tractor [9] phase correction, there are numerous olive ancestry switches (S8 and S12 Figs), which are not expected given that the olive parent is 100% olive (S7 Fig). The presence of these ancestry switches in the F1 individual can be explained by phase switch errors. Importantly, after using Tractor [9] correction, most of these ancestry switches are resolved, with a few exceptions (S9-S11 and S13-S15 Figs). This agrees with what has been reported based on simulated data [9]. Note that there are some chromosome regions where the F1 individual (both with and without Tractor [9] correction) shows olive ancestry in both chromosomes. These regions are also regions where the parental yellow shows olive ancestry [S6 Fig] (although there is also evidence of local ancestry inconsistencies, which will be described in more detail below). 

In the new version of the manuscript, we also explored in more detail the reduction in the number of local ancestry switches for the entire sample after different rounds of RFMix [10] local ancestry estimation and subsequent unkinking using Tractor [9], for the Beagle [8] and SHAPEIT5/IMPUTE5 [6,11] pipelines. We added a paragraph in the ‘Methods’ section from lines 258-265 explaining how we did this. Additionally, lines 393-414 in the ‘Results’ section, and lines 458-471 and a new paragraph at lines 490-505 has been included in the ‘Discussion’ section documenting and explaining our results. Further, we generated 10 new supplemental figures and two new supplemental table (S6-S15 Figs, S5 and S6 Tables) where readers can visualize our findings. Overall, after unkinking the second round of RFMix [10], we observed a reduction in local ancestry switches of nearly 35% (Beagle [8] pipeline) and 23% (SHAPEIT5/IMPUTE5 [6,11] pipeline), with respect to the original RFMix [10] output (S6 Table). We also observed that the number of local ancestry switches is systematically higher with the Beagle [8] pipeline. After unkinking post RFMix [10] round 2, there are 24.5% less local ancestry switches with SHAPEIT5/IMPUTE5 [6,11] than with Beagle [8]. These results can be viewed in S6 Table. 

With respect to local ancestry errors, despite the high genetic differentiation of yellow and olive baboons, the small reference panel used in local ancestry estimation with RFMix [10] can result in local ancestry inaccuracies. We can also take advantage of the F1 hybrid to check for local ancestry inconsistencies: that is, windows where the local ancestry of the F1 shows Mendelian incompatibilities with the parental local ancestry estimates (e.g. when the F1 has two copies of olive or yellow ancestry, and the parents two copies of yellow, or olive ancestry, respectively; in this situation, the F1 should be expected to have one copy of each allele). Exploring this at the level of the entire genome, we see that 3.55% of the markers show local ancestry inconsistencies, which is a relatively small proportion. Typically, when we see inconsistencies, they are observed in windows with a relatively small number of markers (less than 10,000 markers). New data in S5 Table shows the local ancestry information for the trio, highlighting local ancestry inconsistencies after two rounds of RFMix [10] followed by phase correction using Tractor [9].

In summary, both phase switch errors and local ancestry inconsistencies are observed in our sample. Our analyses indicate that the use of Tractor [9] correction fixes many of the phase switch errors, although clearly some of these errors are still present after correction. Also, based on our analysis of the F1 hybrid, there is evidence of local ancestry inconsistencies, although these seem to affect a relatively small proportion of the genome (less than 5% of the markers).    

References

1. Robinson JA, Belsare S, Birnbaum S, Newman DE, Chan J, Glenn JP, et al. Analysis of 100 high-coverage genomes from a pedigreed captive baboon colony. Genome Res. 2019 Mar;29(5):848-56. doi: 10.1101/gr.247122.118 

2. Rogers J, Raveendran M, Harris RA, Mailund T, Leppälä K, Athanasiadis G, et al. The comparative genomics and complex population history of Papio baboons.  Sci Adv. 2019 Jan;5(1):1-14. doi: 10.1126/sciadv.aau6947 

3. Sørensen EF, Harris RA, Zhang L, Raveendran M, Kuderna LFK, Walker JA, et al. Genome-wide coancestry reveals details of ancient and recent male-drive

---

## [Editor Report · Decision Letter 1]

27 May 2024

Global and Local Ancestry Estimation in a Captive Baboon Colony

PONE-D-24-01461R1

Dear Dr. Christopher Jon Richard Kendall,

We’re pleased to inform you that your manuscript has been judged scientifically suitable for publication and will be formally accepted for publication once it meets all outstanding technical requirements.

Kind regards,

Gyaneshwer Chaubey

Academic Editor

PLOS ONE
---

## [Editor Report · Acceptance letter]

29 May 2024

PONE-D-24-01461R1 

PLOS ONE

Dear Dr. Kendall, 

I'm pleased to inform you that your manuscript has been deemed suitable for publication in PLOS ONE. Congratulations! Your manuscript is now being handed over to our production team.

Kind regards, 

on behalf of

Gyaneshwer Chaubey 

Academic Editor

PLOS ONE